# Conceptual Limitations of the Probability Density Function Method for Modeling Turbulent Premixed Combustion and Closed-Form Description of Chemical Reactions' Effects

Vladimir L. Zimont [1,2]

1  CRS4 Research Center POLARIS, 09010 Pula, Italy; zimont@crs4.it
2  N. N. Semenov Federal Research Center for Chemical Physics of the Russian, Academy of Sciences, Kosygina St., 4, 117977 Moscow, Russia

**Abstract:** In this paper, we critically analyzed possibilities of probability density function (PDF) methods for the closed-form description of combustion chemical effects in turbulent premixed flames. We came to the conclusion that the concept of a closed-form description of chemical effects in the classical modeling strategy in the PDF method based on the use of reaction-independent mixing models is not applicable to turbulent flames. The reason for this is the strong dependence of mixing on the combustion reactions due to the thin-reaction-zone nature of turbulent combustion confirmed in recent optical studies and direct numerical simulations. In this case, the chemical effect is caused by coupled reaction–diffusion processes that take place in thin zones of instantaneous combustion. We considered possible alternative modeling strategies in the PDF method that would allow the chemical effects to be described in a closed form and came to the conclusion that this is possible only in a hypothetical case where instantaneous combustion occurs in reaction zones identical to the reaction zone of the undisturbed laminar flame. For turbulent combustion in the laminar flamelet regime, we use an inverse modeling strategy where the model PDF directly contains the characteristics of the laminar flame. For turbulent combustion in the distributed preheat zone regime, we offer an original joint direct/inverse modeling strategy. For turbulent combustion in the thickened flamelet regime, we combine the joint direct/inverse and inverse modeling strategies correspondingly for simulation of the thickened flamelet structure and for the determination of the global characteristics of the turbulent flame.

**Keywords:** turbulent premixed combustion; PDF method; closed-form chemical effects

## 1. Introduction

We begin with a quote from Williams' Hottel Plenary Lecture entitled 'The role of theory in combustion science' [1]: "It is relevant to distinguish between the science and the technology of the subject. The march of technology has never hesitated. It uses science whenever possible but often, especially in combustion, forges ahead by trial and error, or fortuitously by application of scientific misconception, but without scientific understanding". In this paper, we analyze the probability density function (PDF) method in the turbulent combustion theory. We do not consider applications of the PDF method to practical modeling of turbulent combustion ("technology" in Williams' terminology). We consider and discuss a conceptual mistake ("scientific misconception" in Williams' terms) arising in PDF modeling of turbulent combustion, which has caused erroneous interpretation of the mixing terms requiring modeling that appear in the unclosed PDF equations. Although the term "misconception" may seem overly dramatic, its use may be justified in order to emphasize the need for an appropriate analysis of each particular case of doubt.

In this paper, we critically analyze a paradigm for turbulent combustion research in the context of PDF methods based on the use of differential transport equations for

probability density functions of reacting scalars, which has led to the widespread belief that the chemical effects of combustion reactions can be described in a closed form; that is, they do not need to be modeled. We show a misconception in the classical PDF modeling strategy based on this paradigm designed to describe the chemical effects of turbulent combustion in the closed form. The genesis of this misconception is a logical error, which we will show by a critical overview of the statements in some key papers concerning the use of the PDF approach to modeling turbulent combustion.

The quotes from the article "Paradigms in turbulent combustion research" [2] read:

> "While no modelling is required of the reaction term in the PDF equations, a mixing model is needed to account for mixing by molecular diffusion", and "the major issue with mixing models in PDF methods is as follows: there is no explicit coupling between reaction and mixing, although such coupling is implicit through the shape of the PDF"

(slightly edited quote from Reference [2])

These quotes state that the reaction term in the PDF equations is closed and it does not need to be modeled (which is right), while unclosed mixing terms requiring modeling do not depend explicitly on the rate of the combustion reaction, which is wrong, and we will indicate below the physical reason for this error. The genesis of this misconception (the belief in this erroneous issue with mixing models in PDF methods) is caused by the formal interpretation of the following statements:

> "The most notable feature of the transport equation for PDF is that the effects of the reaction appear in closed form: no terms associated with the reaction need to be modelled". "PDF methods appear to be the best approach available since they completely overcome the closure problem associated with nonlinear reaction rates" [3]

and

> "this closure problem simply does not exist: in the exact conservation equation for the composition joint PDF, the effects of the reaction are in closed form (i.e., they do not need to be modelled)" [4]

In other words, "the effects of the reaction appear in closed form" means that the effects of the combustion reaction appear only in the chemical source term that is closed, and do not appear in other terms of the unclosed PDF equations. "PDF methods ... completely overcome the closure problem associated with nonlinear reaction rates" and "no terms associated with the reaction need to be modelled" mean that mixing term requiring modeling does not depend explicitly on the reaction. This false reasoning has led to the belief that the "PDF methods appear to be the best approach available since they completely overcome the closure problem associated with nonlinear reaction rates". In the cited statements, the independence of the molecular mixing terms on the combustion reaction is considered to be a general property of the PDF methods. In this case, PDF equations with closed-form chemical source terms that include reaction-independent mixing models, such as the Interaction by Exchange with the Mean (IEM), Euclidean Minimum Spanning Tree (EMST), and other similar mixing models mentioned in Reference [2], describe all the chemical effects in turbulent combustion in the closed form, that is, without their modeling. This reasoning seems logically flawless, but it is not, as we will show below.

The quote from a more recent review article on the progress in probability density function methods reads:

> "Probability density function (PDF) methods offer compelling advantages for modelling chemically reacting turbulent flows. In particular, they provide an elegant and effective resolution to the closure problems that arise from averaging or filtering the highly nonlinear chemical source terms in the instantaneous governing equations" [5]

Obviously, "an elegant and effective resolution to the closure problems" refers only to the chemical source terms, and "compelling advantages" can only be when molecular mixing processes are independent of the combustion reaction rate, which is implicitly assumed in this quote.

The point is that the reasoning in the above quotes contains a logical error. Formally thinking, since the mixing term of the unclosed PDF equation does not contain the reaction rate, this implies that the molecular mixing rate in turbulent flames does not depend explicitly on the chemical reaction, and hence the chemical effects in turbulent combustion are described in the closed form. This is not true. The dependence of the mixing rate on the combustion reaction is inevitable as the instantaneous gradients of reacting scalars quantifying the mixing intensity directly depend on the coupled reaction–diffusion processes that take place in the instantaneous combustion zones. Hence, the mixing term always depends explicitly on the rate of the combustion reaction. However, this dependence could be expected to be relatively weak in the case of spatially distributed reaction mechanisms of instantaneous combustion in the turbulent premixed flame.

It seems that developing, starting from the seventies of the last century, models of turbulent combustion in which probability density functions play a central role, many researchers believed that in the case of fully developed turbulence there is an entrainment of turbulent eddies of all sizes into the instantaneous reaction zone, which leads to the mechanism of distributed combustion. Hence, within the framework of this belief, the use of mixing models that do not contain chemical reaction rates seemed to be justified.

At that time, in Reference [6], we analyzed the mechanism of turbulent combustion using a Kolmogorov-type hypothesis-based constant-density approach. This theoretical analysis showed the existence of a limit of expansion of the instantaneous flame by the penetration of the turbulent eddies into the preheat zone, that is, the thickened instantaneous flame remains relatively thin and therefore strongly wrinkled. Here, as in the case of the laminar flamelet regime, the classical modeling strategy in the PDF method is not applicable. Experimental data that definitely confirmed this mechanism of turbulent premixed combustion mechanism were obtained much later.

Borghi, analyzing the results obtained in Reference [6], wrote in Reference [7] as follows: "In addition, the need for an intermediate model was emphasized and, more recently, the study of Zimont has allowed one to quantify, in some sense, this new model and identify it as a regime of thickened flamelets, wrinkled again by the largest scales of the turbulence". Borghi proposed a diagram that shows the boundaries between the intermediate thickened flamelet regime, and the laminar flamelet and distributed combustion regimes using the inequalities obtained in Reference [6], which indicates the range of parameters where the thickened flamelet combustion takes place. We consider the concept of the thickened flamelet regime at moderate-to-strong turbulence as a more realistic alternative to the concept of distributed combustion regime assumed in the classical PDF approach.

The inevitable dependence of the mixing terms in the PDF equations on the rate of combustion reactions is associated with the nature of real turbulent flames. In recent years, a considerable number of experimental and direct numerical simulation (DNS) studies have been published that purport to elucidate the structure of instantaneous combustion zones in turbulent flames. The general conclusion of these studies (we briefly review them in the main body of the paper) is that the structures of the combustion zone in turbulent and laminar flames are similar even when premixed flames are subjected to extreme levels of turbulence.

This explains why the seemingly flawless arguments in favor of the independence of the mixing terms in the unclosed PDF equations on the combustion reactions turn out to be wrong. Therefore, the belief in the possibility of a closed-form description of the chemical effects in the framework of the classical modeling strategy described above has led to a dramatic overestimation of the capabilities of the PDF method to give an adequate description of real turbulent flames. Hence, an actual devaluation of the studies

of turbulent flames, which are based on the PDF equations where the mixing models used do not depend on the chemical reactions, seems inevitable.

It should be noted that a significant number of researchers are sure that the mixing term can be directly dependent on the combustion reaction rate. They implicitly disagree with this classical PDF modeling strategy for turbulent flames, and they are right. The results of direct numerical simulations clearly demonstrate the dependence of the mixing processes in turbulent flames on the chemistry; see, for example, Reference [8]. Some researchers tried to address this problem and to develop mixing models including chemical reactions, for example, in the recent papers, References [9,10]. We note that the use of this kind of mixing models reasonable for some applications does not allow for the description of the chemical effects in a closed form. Since our goal is to investigate the possibility of describing the chemical effects in a closed form, we do not consider approaches that used mixing models involving chemical reactions.

Our main claim is that the capabilities of the PDF method to provide an adequate description of the turbulent premixed flame and a closed-form modeling of the chemical effects caused by the combustion reactions are overestimated in the literature. This is not because of some flaws in the PDF method and the codes developed for its application in numerical simulations: research in this area is usually conducted to a very high standard. The reason for the overestimation is that the classical modeling strategy in the PDF method is, in fact, suitable for the spatially distributed combustion only but not for the thin-reaction-zone combustion mechanism common to the turbulent flames in the differently classified regimes; that is, the potential of the classical PDF method does not match the real nature of the turbulent combustion phenomenon.

Nevertheless, we show that the closed-form description of chemical effects caused by fast reactions is possible for a hypothetical model of the turbulent premixed flame where, regardless of the structure of the preheat zone, instantaneous combustion takes place in thin layers whose structure is identical to the structure of the reaction zone of the undisturbed laminar flame. This can be performed in the context of an alternative PDF approach that we describe in this paper. Although for simplicity we consider the PDF equations corresponding to the case of a one-step reaction, the main conclusions are also applicable to the PDF modeling of combustion using detailed chemical kinetics.

The paper is organized as follows:

In Section 2, we briefly consider the experimental and numerical works devoted to the study of instantaneous combustion zones in turbulent flames, the results of which formed the basis of our PDF approach to modeling turbulent combustion.

In Section 3, we qualitatively describe the direct, inverse, and joint direct/inverse modeling strategies in the PDF method that are used in the following sections for the development of model equations intended to describe turbulent premixed flames in different combustion regimes.

In Section 4, we consider turbulent premixed combustion in the laminar flamelet regime. By comparing the "direct" and "inverse" PDF equations, we get the exact expression for the mixing term in this unclosed PDF equation that shows its explicit dependence on the combustion reaction. Moreover, the structure of this expression makes it clear why the inverse equation describing the chemical effect in the closed form does not contain the chemical source term of the unclosed PDF equation.

In Section 5, we describe an original joint direct/indirect modeling strategy in the PDF method for simulating of turbulent flames with the distributed preheat zone and thin reaction zone assumed to be identical to the reaction zone in the laminar flame.

In Section 6, we propose to use the joint direct/indirect strategy for modeling turbulent premixed combustion in the thickened flamelet regime, which leads to the closed-form description of the chemical effects caused by coupled reaction–diffusion processes inside the laminar reaction zone.

Section 7 contains the conclusions.

In Appendix A, we present the derivation of the unclosed PDF equations used in this paper.

## 2. The Structure of the Combustion Zones in Fully Developed Turbulent Flames

The experimental and numerical results obtained in References [11–18] serve as the foundation for our PDF approach to modeling premixed turbulent combustion based on a laminar reaction zone paradigm analogous to the laminar flamelet paradigm in non-premixed turbulent combustion discussed in Reference [2]. Therefore, we pay special attention to these studies not directly related to probabilistic methods and briefly consider them in this section.

A review of studies of the structure of the flamelet in laboratory premixed flames performed in Reference [11] showed that the laminar flamelet regime was observed in the majority of experiments. The thickened flamelet regime, when instantaneous combustion takes place in a flamelet sheet widened by small-scale turbulence and strongly wrinkled by large-scale turbulence, was observed only in several laboratory experiments analyzed in Reference [11]. Of these studies, we especially note Reference [12], where the preheat zone thickening was observed in the Bunsen high-velocity (the speed of the mixture $U = 65$ m/s) stoichiometric methane–air flame using a two-plane, two-dimensional Rayleigh thermometry technique. It was found that this broadening is correlated well with the increase of turbulence intensity: the dimensionless width of the thickened (microturbulent) flamelet $\delta_{mt}$, $\delta_{mt}/\delta_L$, increased from 2.5 to 5 with an increase in the dimensionless velocity fluctuation, $u'/S_L$, from 3 to 20. Here, $\delta_L$ and $S_L$ are the width and speed of the undisturbed laminar flame, $u' = \left(\overline{u'^2}\right)^{1/2}$ is the velocity fluctuation.

The results of optical investigations of the inner structure the of premixed Bunsen flame presented in Reference [13] were obtained for the broad ranges of the dimensionless parameters $u'/S_L, L/\delta_L$, $Re_t$, and $Da$ ($4.5 \leq u'/S_L \leq 246, 15 \leq L/\delta_L \leq 215$, $760 \leq Re_t \leq 99,000$, $0.1 \leq Da \leq 12$) related to the regions of the thickened flamelet and distributed combustion regimes of the Borghi diagram. Here, $Re_t = u'L/\nu$ is the turbulent Reynolds number, where $L$ is the integral length scale of turbulence and $\nu$ is the kinematical viscosity coefficient; $Da = \tau_t/\tau_{ch}$ is the Damköhler number, where $\tau_t = L/u'$ is the turbulent time and $\tau_{ch}$ is the chemical time. Combustion occurred in the wrinkled reaction zones with a thickness close to the width of the reaction zone of the laminar flame, including the flames with very broad preheat zones.

The authors of Reference [13] conducted a special study in Reference [14] to explain this phenomenon, which seemed to them obscure. They wrote [14]: "The authors previously found that very broad preheat layers were achieved for turbulence levels $u'/S_L$ up to 243. Surprisingly, the reaction layer thickness did not increase, despite having Kolmogorov scales smaller than the laminar reaction layer thickness". They performed an experimental study of the preheat zone using optical methods: where fluorescence imaging identified the reaction zone boundary and particle image velocimetry diagnostics were applied simultaneously. "Results indicate that the turbulence level does not decrease within the broad preheat layers". At the same time, "the integral scale increased by 50% across the preheat layer". "One explanation for this result is that small eddies are dissipated in the preheat zone".

We notice that a similar phenomenon exists in the case of non-premixed turbulent combustion. Experimental data obtained in non-premixed flames under conditions of strong turbulence [15] showed that instantaneous combustion took place in the laminar flamelet and thickened flamelets were not observed, which corresponded to the laminar flamelet paradigm in non-premixed turbulent combustion [2].

Flame thicknesses of the preheat and heat release layers of the turbulent $CH_4/H_2/$air premixed Bunsen flames adopted with three hydrogen fractions of 0%, 30%, and 60% were measured in Reference [16] using the optical methods. The preheat zone thickness increased to about 3–6 times compared to the laminar preheat thickness. An apparently

decreased preheat zone thickness with hydrogen addition is observed. The mean heat release thickness is nearly not affected by the turbulence or hydrogen addition.

Similar results were obtained from direct numerical simulations even in the cases of very small Damköhler numbers. For example, the recent result of DNSs of constant-density turbulent premixed flame propagation obtained in the range of Damköhler numbers, $0.01 \leq Da < 1$, which corresponds to the distributed combustion regime in the Borghi diagram, "shows that the reaction zone thickness is statistically close to the thickness of the reaction zone in a laminar flow" [17]. Three-dimensional DNSs of the statistically planar steady-state turbulent flames have been used in Reference [18] in an attempt to produce distributed burning in extremely low Damköhler number lean methane and hydrogen flames ($0.00235 \leq Da \leq 0.044$ for both fuels). "Turbulence has to survive deeper into the methane flame to disrupt the reaction zone" [18], which is consistent with the result obtained in Reference [17]. "Dilatation across the flame means that extremely large Karlovitz numbers are required; even at the extreme levels of turbulence studied (up to a Karlovitz number of 8,767), distributed burning was only achieved in the hydrogen case. In this case, turbulence was found to broaden the reaction zone visually by around an order of magnitude". The results obtained in Reference [18] show that the tendency towards a distributed burning regime took place only for hydrogen and with turbulent parameters that are not realized in experimental flames.

Summing up, we can say that these and other relatively recent experimental and DNS results have argued in favor of the laminar-like nature of the thin reaction zones in fully developed turbulent flames, that is, the laminar-reaction-zone paradigm seems relatively well established. This leads to two conclusions:

1.  The PDF approaches, based on the use of reaction-independent mixing models, do not correspond to the nature of the turbulent premixed flame due to strong coupling of the mixing and chemical processes in the thin combustion zones.
2.  In the alternative PDF approaches under consideration, we apply the laminar-reaction-zone paradigm and assume that the structure of the reaction zone in the turbulent premixed flame is identical to the structure of the reaction zone in the laminar premixed flame.

## 3. Direct, Inverse, and Joint Direct/Inverse Modeling Strategies in the PDF Method

In this section, we qualitatively describe modeling strategies in the PDF method that are used in the following sections for the development of model PDF providing the closed-form description of chemical effects in turbulent premixed flames in different regimes of combustion.

We distinguish three modeling strategies in the PDF method:

-   A direct strategy, where the problem reduces to modeling the unclosed mixing term. The resulting model PDF equations containing closed chemical source terms describe, along with the global characteristics of the turbulent flame, its internal structure, which is statistically described by probabilistic functions. Therefore, these model PDF equations are also intended to predict combustion regimes. Hence, the logical diagram of this strategy can be represented as follows:

$$\text{micro} - \text{mixing model} \Rightarrow \text{modelPDF equation} \Rightarrow \text{combustion regime}.$$

-   An inverse modeling strategy, starting from an analysis of the combustion regime and a subsequent theoretical or numerical analysis of the structure of the instantaneous flame that is determined by the coupled transport and chemical processes. The model PDF equations explicitly contain the characteristics of the instantaneous flame and at the same time do not contain the closed chemical source terms appearing in the unclosed PDF equations. The logical diagram of this strategy is as follows:

$$\text{combustion regime} \Rightarrow \text{instantaneousflame structure} \Rightarrow \text{model PDF equation}.$$

- A joint direct/inverse modeling strategy, where the preheat zone is described using the framework of the direct strategy, while the instantaneous reaction zone is described in the context of the inverse strategy. In this case, the chemical effects caused by the coupled reaction–diffusion processes in the thin combustion zones can be described in the closed form.

We think that these names of the modeling strategies reflect the essence of the matter.

The direct modeling strategy in the classic formulation (where "there is no explicit coupling between reaction and mixing" [2]) yielding the closed form of the chemical effects, does not provide adequate estimation of the characteristics associated with fast reactions (the heat release rate, prompt NO formation, and so on) due to strong coupling of the mixing and chemical processes in the thin combustion zones observed in the experiments and direct numerical simulations.

The inverse modeling strategy is known in the literature. It was used to formulate the PDF equation for modeling premixed turbulent combustion in the laminar flamelet regime. Along with this case, we use the inverse strategy for analyzing more complex problems of PDF modeling of turbulent premixed combustion. Formulating the PDF equations in the context of the joint direct/inverse modeling strategy, we assume that the effect of the reaction in the turbulent preheat zone is negligible and that the thin reaction zone is laminar. This makes it possible to use a mixing model independent of the chemical reaction, and the results for the reaction zone obtained in the theory of laminar flame. We formulate the PDF equations of the direct and inverse subproblems for turbulent premixed combustion in the distributed preheat zone regime. We also consider the possibility of applying the joint direct/inverse modeling strategy to study the structure of the thickened flamelet and obtain the flamelet characteristics necessary to form an inverse PDF equation describing the turbulent flame in the thickened flamelet regime.

## 4. PDF Modeling of Turbulent Combustion in the Laminar Flamelet Regime

We formulate the PDF equation using the reaction progress variable $c$ ($c = 0$ and $c = 1$ in the unburned and burned gases). Instantaneous premixed combustion is described by the following equation

$$\partial(\rho c)/\partial t + \nabla \cdot (\rho \vec{u} c) = \nabla(\rho D \nabla c) + \rho W, \tag{1}$$

where $\rho$ is the density, $D$ is the molecular diffusivity ($D \sim \nu$) and $W(c)$ the reaction rate per unit mass. Corresponding unclosed equation in terms of PDF $p(c; \vec{x}, t)$ (denote in the equation by $p$ for brevity) in the coordinate form is as follows (see Equation (A10) in the Appendix A):

$$\partial(\rho p)/\partial t + \partial(\rho(\overline{u}_k)_c p)/\partial x_k + \partial^2(\rho \overline{E}_c p)/\partial c^2 + \partial(\rho p W)/\partial c = 0, \tag{2}$$

where $(\overline{u}_k)_c$ is the conditional mean velocity and $\overline{E}_c = \overline{D(\partial c/\partial x_k \cdot \partial c/\partial x_k)}_c$ is the conditional mean dissipation rate of the progress variable. In this equation the second term is the convective term, the third term is the unclosed (requiring modeling) mixing term, and the fourth term is the closed chemical source term. For reader convenince we present in Appendix A the derivation of the used in this paper unclosed PDF equations. (We did not consider equations for the joint velocity-scalar PDF because our goal was to analyze the possibility of describing chemical effects in a closed form.)

The use of mixing models developed in the context of "the major issue with mixing models in PDF methods" for example, models IEM, MS, LMSE and EMST mentioned in Reference [2], where the conditional micromixing intensity $\overline{E}_c$ does not depend on the reaction rate, leads to model PDF equations that describe the chemical effects in the closed form. Due to the lack of the coupling between chemical reaction and molecular mixing, these model PDF equations describe a kinetic regime of combustion. Obviously, these

models are not applicable in the laminar flamelet regime, where the instantaneous flame is described by the following kinematic equation:

$$\partial(\rho c)/\partial t + \nabla \cdot (\rho \vec{u} c) = \rho_u S_L \left| \nabla c \right|, \tag{3}$$

where $S_L$ is the speed of the flame relative to the unburned gas with the density $\rho_u$. Comparing of Equations (1) and (3) leads to the formula

$$\rho_u S_L |\nabla c| = \nabla(\rho D \nabla c) + \rho W. \tag{4}$$

Equation (4) shows that the source term is determined by molecular diffusion and chemical reaction.

For a better understanding of our further statements, we emphasize that it would be a logical mistake to assert the independence of diffusion and chemical processes in the instantaneous laminar flame from the fact that diffusion and reaction in Equation (4) are described by separate terms. The speed $S_L$ and the progress variable gradient $|\nabla c| = dc/dn$, where $n-$ axis is normal to the instantaneous flamelet, are determined by coupled reaction-diffusion processes that occur inside the instantaneous laminar flame. The gradient $|\nabla c|(n)$ can be represented using $c(n)$ as a function of the progress variable $|\nabla c| = f_L(c)$. The speed $S_L$ and function $f_L(c)$ are assumed to be known from the theory of the laminar flame. We note that the speed $S_L$ is strictly defined only for the one-dimensional laminar flame. The use this speed in Equation (3) is an approximation that is justified when $\delta_L << \eta$, where $\delta_L \sim (D/\tau_{ch})^{1/2}$ is the width of laminar flame and $\eta = L\mathrm{Re}_t^{-3/4}$ is the Kilmogorov microscale of turbulence equal in order of magnitude to the minimum size of turbulent eddies.

The PDF equation corresponding to the kinematic Equation (3) is as follows (see Equation (A15) in the Appendix A):

$$\partial(\rho p)/\partial t + \partial(\rho(\overline{u}_k)_c p)/\partial x_k + \rho_u S_L \partial(f_L p)/\partial c = 0. \tag{5}$$

This equation describes the chemical effect in the closed form (this result is known in the literature [5,19]), as the speed $S_L$ and the function $f_L(c)$ defined inside the one-dimensional flame can be obtained in the theory of the laminar flame without modeling. For simplification of the problem, we can assuming that the speed $S_L$ is a known physicochemical characteristic of the combustible mixture, and estimate the characteristic value of $f_L$ using the expression $f_L \approx 1/\delta_L \approx S_L/D$ (as $\delta_L \approx D/S_L$ that follows from $S_L \approx (D/\tau_{ch})^{1/2}$ and $\delta_L \approx (D\tau_{ch})^{1/2}$).

It is significant that in the case of using detailed chemical kinetics, this inverse approach allows one to describe in a closed form all the chemical effects associated with fast reaction, for example, prompt-NO formation. For this, we must use the kinematical equation analogous to Equation (3), which is written in terms of the prompt-NO species $c_{NO}$, we get the PDF equation in terms of $p(c_{NO})$ similar to Equation (3). In this equation intead of $f_L(c)$ would be $f_L^{\bullet}(c_{NO}) = |\nabla c_{NO}|(c_{NO})$ following from the functions $c_{NO}(n)$ and $|\nabla c_{NO}|(n)$, with must be calculated using the result of simulation with detailed chemistry of the laminar flame

A comparison of the Equations (2) and (5) shows that the unclosed mixing term of the Equation (1) $\partial^2(\rho \overline{E}_c p)/\partial c^2$ obeys the expression

$$\partial^2(\rho \overline{E}_c p)/\partial c^2 = \rho_u S_L \partial(f_L p)/\partial c - \partial(\rho p W)/\partial c. \tag{6}$$

The Equation (6) clearly shows that the mixing term explicitly depends on the reaction rate $W(c)$. The only way to describe in the context of the direct modeling strategy the chemical effects of turbulent combustion in a closed form is to use the mixing term described by the formula (6), which explicitly contains information about the structure and speed of the instantaneous flame ($f_L(c)$ and $S_L$), and summand $-\partial(\rho p W)/\partial c$. The substitution of

Equation (6) into Equation (2) annihilates the closed chemical source $\partial(\rho p W)/\partial c$ of this unclosed equation. This means that in the progress-variable approach, the only PDF equation that adequately describes in the closed form the chemical effect of turbulent premixed combustion in the laminar flamelet regime is the unclosed Equation (5) formulated in the context of the inverse modelling strategy. In principle, we can get a description of chemical effects in a closed form also in the context of a direct modelling strategy. But for this we have to use mixing models that do not depend on the combustion reaction. It is obvious that in this case the regime of combustion cannot be laminar flamelet.

We formulate a model inverse strategy equation in term of the Favre probability density function $\widetilde{p}(c; \vec{x}, t) = \overline{\rho p(c; \vec{x}, t)}/\overline{\rho}$ using the unclosed equation corresponding to Equation (2) that is as follows (see Equation (A17) in the Appendix A):

$$\overline{\rho} \cdot \partial \widetilde{p}/\partial t + \overline{\rho} \widetilde{\vec{u}} \nabla \widetilde{p} + \nabla \cdot [(\overline{\rho \vec{u}''})_c \widetilde{p}] + \rho_u S_L \partial[(f_L/\rho)\overline{\rho}\widetilde{p}]/\partial c = 0, \tag{7}$$

where $\overline{\rho}$ is the mean density, $\widetilde{\vec{u}} = \overline{\rho \vec{u}}/\overline{\rho}$ is the Favre-averaged velocity, $\vec{u}'' = \vec{u} - \widetilde{\vec{u}}$ is the Favre velocity fluctuation $\vec{u}'' = \vec{u} - \widetilde{\vec{u}}$ and subscript $c$ means conditional averaging. Using the gradient form for the transport terms $\nabla \cdot [(\overline{\rho \vec{u}''})_c \widetilde{p}] = -\nabla(\overline{\rho} D_t \nabla \widetilde{p})$, where $D_t \approx u'L$ is the turbulent diffusion coefficient, we obtain the following model equation describing turbulent premixed combustion in the laminar flamelet regime

$$\overline{\rho} \cdot \partial \widetilde{p}/\partial t + \overline{\rho} \widetilde{\vec{u}} \nabla \widetilde{p} - \nabla(\overline{\rho} D_t \nabla \widetilde{p}) + \rho_u S_L \partial[(f_L/\rho)\overline{\rho}\widetilde{p}]/\partial c = 0. \tag{8}$$

The author calls the reader's attention to, at first glance, paradoxical result that the PDF Equations (5), (7) and (8) formulated in the context of the inverse modeling strategy does not contain the closed chemical source, which is inherent term of model PDF equations developed in the context of the direct modeling strategy. Clear explanation of this fact is important for understanding the issue underdiscussion. The explanation for this seeming paradox is that the chemical effects in the turbulent flame are caused by coupled diffusion-reaction processes that occur in a thin zone of instantaneous combustion. The reader should not be confused by the fact that diffusion and combustion are described in Equation (4) by different terms. This, when formally considered, may lead to the erroneous belief that diffusion processes in a flame occur independently. Similarly, from the fact that the mixing term does not contain $W(c)$, it does not follow that this term does not explicitly depend on the combustion reaction, as we have shown above.

Hence, the direct modelling strategy in the PDF method with the mentioned above classical issue with mixing models (independent of the combustion reaction mixing models) is not suitable for description of turbulent combustion in the laminar flamelet regime. It is obvious, that possible mixing models depending explicitly on the reaction rate, which do not contain the structure of the instantaneous flame, cannot adequately describe the turbulent flame in the laminar combustion regime. Furthermore, these models cannot describe the chemical effects in the closed form.

These conclusions remain valid for the thickened flamelet regime and classified as the distributed combustion regime of the turbulent due to above-mentioned thin-reaction-zone nature of real turbulent premixed combustion. The reason is that the chemical effects in turbulent flames are caused by coupled diffusion-reaction processes taking place in the laminar-type instantaneous combustion zones.

In the next section we will consider the distributed combustion regime observed in [13,14] in premixed flames subjected to extreme levels of turbulence, and then, in the following section, we will study the possibilities of the PDF method for adequate modeling of the turbulent flame in the thickened flamelet regime, which is intermediate between the laminar flamelet regime (at relatively weak turbulence) and distributed preheat zone regime (at very strong turbulence).

## 5. The Distributed Preheat Zone Regime and a Joint Direct/Inverse Modeling Strategy

The direct and inverse PDF modeling strategies are not immediately applicable to the turbulent flames subjected by extremely large level of turbulence. The point is that mentioned above recent experimental studies of the premixed flames using optical methods [13,14] shows that in this case only the preheat zones occurred distributed, while the observed instantaneous laminar-like reaction zones remained thin. This means that the "the major issue with mixing models in PDF methods" is not justified in this case. At the same time, the instantaneous flame does not have a clearly defined structure, which is a condition for the applicability of the inverse PDF modeling strategy. To overcome this impediment, we propose a joint direct/inverse strategy in the PDF method.

In the joint direct/inverse modeling strategy, the preheat zone is descried in the context of the direct strategy. The ability to neglect the effect of the combustion reaction in the preheat zone allowed us to omit the chemical source term and use a mixing model that does not depend on the combustion reaction. At the same time, the instantaneous reaction layer is described in the context of the inverse modeling strategy, assuming the structure of the laminar-type combustion reaction layer is known. This equation also does not contain the close-form chemical source term.

As an example to illustrate this idea (that can be turned into amodelingproblem), we formulate a joint direct/inverse model PDF equation in terms of the Favre PDF $\widetilde{p}(c; \overrightarrow{x}, t)$. To do this, we first separately obtain the equations for the distributed preheat zone and the laminar-type layer of instantaneous combustion, and then using these equations, we formulate a general equation describing premixed combustion under conditions of very strong turbulence.

The basic unclosed equations containing the mixing and chemical source terms (see Equation (A15) in the Appendix A) is as follows:

$$\overline{\rho} \cdot \partial \widetilde{p} / \partial t + \overline{\rho} \widetilde{\overrightarrow{u}} \nabla \widetilde{p} + \nabla \cdot [(\overline{\rho \overrightarrow{u''}})_c \widetilde{p}] + \overline{\rho} \cdot \partial (\overline{E}_c \widetilde{p}) / \partial c + \partial (W \overline{\rho} \widetilde{p}) / \partial c = 0. \tag{9}$$

We will use in an example below the same as in Equation (8) gradient approximation for the transport term and the Interaction by Exchange with the Mean (IEM) mixing model for approximation of the conditional mean dissipation rate of the progress variable $\overline{E}_c$. This earliest mixing model was proposed in 1960 by Vladimir Frost [20] and then developed independently in Reference [21]. In this model the conditional dissipation rate is approximated by the formula $\overline{E}_c = \omega_c(c - \widetilde{c})$, where $\omega_c$ is the conditional mixing frequency. Assuming that the mixing frequency is the same for all $c$ and equal to $\omega \sim 1/\tau_t$, where $\tau_t \sim L/u'$ is the turbulent time, we obtain the following model equation valid only inside the preheat zone:

$$\overline{\rho} \cdot \partial \widetilde{p} / \partial t + \overline{\rho} \widetilde{\overrightarrow{u}} \nabla \widetilde{p} - \nabla (\overline{\rho} D_t \nabla \overrightarrow{p}) + \overline{\rho} \omega \cdot \partial [(c - \overrightarrow{c}) \widetilde{p}] / \partial c = 0. \tag{10}$$

In order to obtain a PDF equation that is valid in the instantaneous reaction layer, we note that laminar-like reaction zone is adjacent to the burned gas where $c = 1$ and the density $\rho_b$. This reaction zone moves at the speed $S_r = (\rho_u / \rho_b) S_L$ relative to the burnt gas. In this case the source term $\rho_u S_L \partial [(f_L / \rho) \overline{\rho} \widetilde{p}] / \partial c$ in Equaion (8), which was obtained for the case of the laminar flamelet regime, becomes $\rho_b S_r \partial [(f_r / \rho) \overline{\rho} \widetilde{p}] / \partial c$, where $|\nabla c|_r = f_r(c)$ is defined inside the reaction zone . . . ... Hence, an equation governing the PDF $\widetilde{p}(c; \overrightarrow{x}, t)$ inside the instantaneous combustion layer (and not valid outside it) is as follows:

$$\overline{\rho} \cdot \partial \widetilde{p} / \partial t + \overline{\rho} \widetilde{\overrightarrow{u}} \nabla \widetilde{p} + \rho_b S_r \partial [(f_r / \rho) \overline{\rho} \widetilde{p}] / \partial c = \nabla (\overline{\rho} D_t \nabla \overrightarrow{p}). \tag{11}$$

Now we formulate a joint strategy PDF equation using the Heaviside step function $H(x)$, which coincides with Equation (10) in the preheat zone and with Equation (11) in the reaction layer, as follows:

$$\bar{\rho}\cdot\partial\tilde{p}/\partial t + \bar{\rho}\overrightarrow{\tilde{u}}\nabla\tilde{p} - \nabla(\bar{\rho}D_t\nabla\tilde{p}) + \underbrace{(1 - H(c - c_\bullet))\bar{\rho}\omega\cdot\partial[(c - \tilde{c})\tilde{p}]/\partial c}_{1} +$$

$$+ \underbrace{H(c - c_\bullet)\rho_b S_r\partial[(f_r/\rho)\bar{\rho}\tilde{p}]/\partial c}_{2} = 0, \tag{12}$$

where $c_*$ in the Heaviside step function $H(c - c_\bullet)(H(x < 0) = 0, H(x \geq 0) = 1)$ is the progress variable at the conditional border that divides preheat and reaction zones. This made it possible to separately apply the direct and invese modeling strategies to the preheat zone $(0 < c < c_*)$ where the term $1 \neq 0$ and term $2 = 0$, and to the reaction zone $(c_* < c < 1)$ where the term $1 = 0$ and term $2 \neq 0$, that is, Equation (12) has been formulated in the context of the joint direct/inverse modeling strategy. We notice that Equation (12) does not contain closed chemical source.

The speed $S_r$ of propagation of the reaction zone relative the burned gas with the density $\rho_b$ and the function $f_r(c) = |\nabla c|(c)$ inside the reaction layer assumed to be laminar and the value $c_*$ can be estimated via the analysis of the reaction zone in the laminar flame.

The Equation (12) for the Favre PDF of the progress variable describes in the closed form the chemical effects, which are caused by coupled reaction-diffusion processes that take place inside the thin layer where instantaneous combustion takes place. In the case of using detailed kinetics, equations for corresponding species concentration PDF, which can be formulated similar to Equation (12) in the context of the joint direct/inverse modeling strategy, describe in the closed form chemical effects caused by fast chemical reactions that takes place in the instantaneous combustion zone, for example, prompt-NO formation. At the same time, the PDF equations formulated in the context of the classical direct strategy, that is, using independent on reactions mixing models and keeping in them the closed chemical sources, can describe in a closed form chemical effects caused by slow reactions, for example, thermal-NO formation and slow CO oxidation.

## 6. The PDF Method and Thickened Flamelet Regime of Turbulent Premixed Combustion

The thickened flamelet regime of the turbulent premixed flame where instantaneous combustion takes place in strongly wrinkled microturbulent flame takes place when (see the inequalities (1.10) and (1.11) in Reference [6] and (7a) and (7b) in Reference [22])

$$Da^{1/2} >> 1 >> Da^{3/2}\text{Re}_t^{-3/4}. \tag{13}$$

These inequalities take place in the case of large Reynolds numbers and moderately large Damköhler numbers, which is typical for the large-scale industrial premixed burners. For illustration, we put $\text{Re}_t = 10^3$ and $Da = 10$ (assuming, for example, $u' = 4$ m/s, $L = 0.5$ cm, $\nu = D = 0.2$ cm$^2$/sec and $S_L = 0.4$ m/s) where the inequalities (13) become 3>>1>>0.2.

The speed $S_{mt}$ and width $\delta_{mt}$ of the instantaneous thickened microturbulent flame, the velocity fluctuation $u'_{mt}$, integral scale $L_{mt}$ and the microturbulent diffusion coefficient $D_{mt} \approx u'_{mt}L_{mt}$ in the thickened flamelet are defined by the inertial-range turbulent structures controlled by the turbulent energy dissipation rate $\varepsilon \approx u'^3/L$ and the combustion reaction characterized by the chemical time $\tau_{ch}$. Hence, the dimensional analysis leads to the following expressions:

$$S_{mt} \sim u'_{mt} \sim (\varepsilon\tau_{ch})^{1/2}(a), \ \delta_{mt} \sim L_{mt} \sim \varepsilon^{1/2}\tau_{ch}^{3/2}(b), \ D_{mt} \sim \varepsilon\tau_{ch}^2(c). \tag{14}$$

From Equation (14a) and (14b) it follows that the Damköhler number inside the thickened flamelet $Da_{mt} = \tau_{mt}/\tau_{ch}$ ($\tau_{mt} = L_{mt}/u'_{mt}$ is the turbulent time in the micro-

turbulent flamelet) is equal to $Da_{mt} \approx 1$. The formulas (14) expressed in the term of the Damköhler number are as follows (see Equations (1.6) in Reference [6] and Equations (3) in Reference [22]):

$$
\begin{aligned}
S_{mt} &\approx u'Da^{-1/2} \approx u'_{mt}(a), \\
\delta_{mt} &\approx L \cdot Da^{-3/2} \approx L_{mt}(b), \\
D_{mt} &\approx D_t Da^{-2}(D_t \approx u'L)(c).
\end{aligned} \tag{15}
$$

We assumed that the instantaneous reaction zone is laminar and identical to the reaction zone in a laminar flame, that is, the chemical time $\tau_{ch}$ is the same in the thickened flamelet and undisturbed laminar flame, $\tau_{ch} = D/S_L^2$. Using Equations (15) and the formulas for dimensional speed and widths

$$
\begin{aligned}
S_{mt}/S_L &\approx \delta_{mt}/\delta_L \approx Da^{-1}\mathrm{Re}_t^{1/2}(a), \\
\delta_{mt}/\eta &= Da^{-3/2}\mathrm{Re}_t^{3/4}(b), \\
\delta_L/\eta &= Da^{-1/2}\mathrm{Re}_t^{1/4}(c),
\end{aligned} \tag{16}
$$

we get the numerical estimates for the used above values $\mathrm{Re}_t = 10^3$ and $Da = 10 : \delta_{mt}/L \approx 0.03, S_{mt}/u' \approx 0.3, \delta_{mt}/\delta_L = S_{mt}/S_L \approx 3, \delta_{mt}/\eta \approx 6, \delta_L/\eta \approx 2, D_{mt}/D \approx 10, D_{mt}/D_t \approx 10^{-2}$. The inequalities $\delta_L/\eta > 1$ and $\delta_{mt}/\eta >> 1$ correspond to concept of the microturbulent flamelet where the thickened is caused by small-scale turbulence, and the inequality $D_{mt}/D >> 1$ shows that the contribution of the molecular diffusion into the transport in the microturbulent flamelet is negligible. In this example, $\delta_{mt}/\delta_L \approx 3$, where the width $\delta_L$ is small ($\delta_L \approx D/S_L \sim 0.1$ mm for $D = 0.2$ m$^2$/s and $S_L = 0.4$ m/s). Therefore, the flamelet remains relatively thin, $\delta_{mt}/L \approx 0.03$, and hence strongly wrinkled.

We proposed in [6] the following explanation for this phenomenon. In the instantaneous flame, there is an increase in the progress variable from $c = 0$ to $c = 1$, and hence the characteristic value of its gradient in the thickened flamelet is equal to $\Delta c/\delta_{mt}$, where the increment of the progress variable $\Delta c = 1$. The convective flux $q_c^{mt} = S_{mt} \approx u'Da^{-1/2}$, the microdifsusion flux $q_d^{mt} \approx D_{mt}(\Delta c/\delta_{mt}) \approx u'Da^{-1/2}$ and the chemical transformation rate per unit area of the thickened flamelet $q_{ch}^{mt} \approx (1/t_r^{mt})\delta_{mt} \approx u'Da^{-1/2}$ ($t_r^{mt} = \delta_{mt}/S_{mt}$ is the residence time in the flamelet), that is, of the same order of magnitude (see Equation (1.7) in [6]). Despite the continuous spectrum of turbulent energy at high Reynolds numbers, additional thickening of the instantaneous microturbulent flame does not occur, since this would upset statistical equilibrium between these processes.

As the instantaneous reaction layer in the microturbulent flamelet is assumed to be laminar, its propagation speed $S_r$ inside the flamelet is equal to $S_r = S_L$, and hence $S_{mt} = S_L(\overline{A}/A_0)_r^{mt}$, where $(\overline{A}/A_0)_r^{mt}$ is the mean dimensionless area of the reaction layer inside the miscoturbulent flamelet. Using Equation (15a) and the right inequality in (13), we get the expression $(\overline{A}/A_0)_r^{mt} \approx Da^{-1}\mathrm{Re}_t^{1/2} >> 1$ ($(\overline{A}/A_0)_r^{mt} \approx 3$ for $\mathrm{Re}_t = 10^3$ and $Da = 10$ used above), which shows strong wrinkling of the reaction layer. This estimate indicates that the fluctuations of the progress variable inside the flamelet are significant. Therefore, the quasi-laminar models for the thickened flamelet can be used in the PDF approach only as a first approximation.

For an accurate description of the PDF $\widetilde{p}(\hat{c}; \vec{x}, t)$, we must take into account both contributions caused by the random movement of the thickened flamelet sheet inside the turbulent premixed flame and progress variable fluctuations inside the flamelet. The PDF $\widetilde{p}(\hat{c}; \vec{x}, t)$ caused by the random movement is described by the following inverse PDF equation formulated for the quasi-laminar interpretation of the microturbulent flamelet:

$$
\overline{\rho} \cdot \partial \widetilde{p}/\partial t + \overline{\rho} \widetilde{\vec{u}} \nabla \widetilde{p} - \nabla(\overline{\rho} D_t \nabla \widetilde{p}) + \rho_u S_{mt} \partial[(f_{mt}/\rho)\overline{\rho}\widetilde{p}]/\partial c = 0 \tag{17}
$$

In this equation (an analogue of Equation (8) for the case of the laminar flamelet regime), $S_{mt}$ is the speed of the thickened flamelet relative to the unburned gas with the density $\rho_u$, $f_{mt}(\hat{c}) = |\nabla \hat{c}|(\hat{c})$, where $\hat{c}$ is the local mean progress variable inside the thick-

ened microtuebulent flamelet and $|\nabla \hat{c}| = d\hat{c}/dn$. The characteristic value of the function $f_{mt}$ using Equation (14b) is $f_{mt} \approx 1/\delta_{mt} \approx Da^{3/2}/L$.

The PDF $\widetilde{p}(c; \overrightarrow{x}, t)$ is defined by the PDF $\widetilde{p}(\hat{c}; \overrightarrow{x}, t)$ described by Equation (17) and the conditional probability density functions $\widetilde{p}(c|\hat{c})$ that correspond to all values of the mean progress variable of the profile $\hat{c} = \hat{c}(n)$ inside the thickened flamelet ($0 \leq \hat{c} \leq 1$). The effects of these two processes are described by the joint probability density function $\widetilde{p}(c, \hat{c}; \overrightarrow{x}, t)$, which can be represented as follows:

$$\widetilde{p}(c, \hat{c}; \overrightarrow{x}, t) = \widetilde{p}(c|\hat{c})\widetilde{p}(\hat{c}; \overrightarrow{x}, t) \tag{18}$$

Equation (18), after integration over $\hat{c}$, leads to the formula for the PDF $\widetilde{p}(c; \overrightarrow{x}, t)$:

$$\widetilde{p}(c; \overrightarrow{x}, t) = \int_0^1 \widetilde{p}(c|\hat{c})\widetilde{p}(\hat{c}; \overrightarrow{x}, t) \, d\hat{c} \tag{19}$$

which shows that, to calculate the desired PDF $\widetilde{p}(c; \overrightarrow{x}, t)$, we have to solve Equation (17) to find $\widetilde{p}(\hat{c}; \overrightarrow{x}, t)$ and use the PDF $\widetilde{p}(c|\hat{c})$ defined inside the thickened flamelet. Thus, in order to resolve this problem, we must first investigate the structure and parameters of the microturbulent flamelet. Unlike the case of the laminar flamelet regime, the study of the structure and parameters of the microturbulent flamelet defining the function $f_{mt}(\hat{c}) = |\nabla \hat{c}|(\hat{c})$ used in Equation (17) and PDF $\widetilde{p}(c|\hat{c})$ used in Equation (19) addresses a specific issue in the turbulent combustion theory.

To the author's knowledge, the problem of the thickened flamelet modeling in the framework of the PDF method has not been considered in the literature. Here, we only outline the possible formulation of this problem. We propose, as a possible strategy for PDF modeling of the microturbulent flamelet, to consider the joint direct/indirect PDF equation in terms of $\widetilde{p}(c; n)$ defined inside the thickened flamelet (an analogous of Equation (12)). For this equation, a specific mixing model must be developed which, in the case of constant density, provides results consistent with the formulas (15) obtained in the framework of a hypothesis-based approach [6,22]. Solving this equation would give the functions $\widetilde{p}(c; n)$, $\hat{c}(n)$ and $d\hat{c}/dn = |\nabla \hat{c}|$, and from here, $S_{mt}$, $\widetilde{p}(c|\hat{c})$ and $f_{mt}(\hat{c}) = |\nabla \hat{c}|(\hat{c})$ are determined, which are used in Equations (17) and (19).

We did not study this problem in detail, because, as we indicated in the Section 1, we did not consider the issues of practical modeling, but only analyzed the conceptual difficulties of using the PDF method in the theory of turbulent combustion. But we are sure that the turbulent premixed flame in the thickened flamelet regime is a good example, along with the case of the laminar flamelet regime, for illustrating some problems and misconceptions related to the application of the PDF method in the theory of turbulent combustion. The case of the thickened flamelet regime turns out to be much more difficult to analyze than the case of the laminar flamelet regime.

On the one hand, the classical modeling strategy that would lead to a closed form of chemical effects by using mixing models independent of the combustion reaction is not applicable also in the case of a thickened flamelet regime. This follows from the expression for the mixing term of the unclosed PDF equation:

$$\overline{\rho} \cdot \partial(\overline{E}_c \widetilde{p})/\partial c = \rho_u S_{mt} \partial[(f_{mt}/\rho)\overline{\rho}\widetilde{p}]/\partial c - \partial(W\overline{\rho}\widetilde{p})/\partial c \tag{20}$$

which clearly shows that it depends on the reaction rate. Equation (20) follows from a comparison of Equation (17) and unclosed PDF Equation (9), in which the unknown transport term $\nabla \cdot [(\overline{\rho \overrightarrow{u''}})_c \widetilde{p}]$ is approximated with $-\nabla(\overline{\rho} D_t \nabla \widetilde{p})$.

On the other hand, the reverse modeling strategy is also applicable to the thickened flamelet regime, since there is a clearly expressed structure of instantaneous microturbulent flame. But we first have to know the global and statistical characteristics of the instanta-

neous microturbulent flame necessary for the inverse problem. This is a particular problem of the theory of turbulent combustion (in contrast to the case of the laminar flamelet regime, as the necessary characteristics of the laminar flamelet appearing in the inverse PDF Equations (5) and (7) could be found from the theory of the laminar premixed flame).

## 7. Conclusions

In this study, we analyzed the potentialities and limitations of the PDF method and possible misconceptions in its use for the modeling of turbulent premixed combustion, which was initiated to some extent by Williams' statement cited at the beginning of the Section 1, prompting the author to take a critical look at the problem, and leading to the following conclusions:

Conclusion 1: Of the two major issues for the probabilistic modeling paradigm of turbulent combustion formulated in Reference [2], "no modelling is required of the reaction term in the PDF equations" and "no explicit coupling between reaction and mixing", the former is correct due to the closed form of the source term in the unclosed PDF equations, while the latter as a general statement is erroneous. It is based on a logical category mistake, which consists in the following: the absence of reaction rates in the unknown mixing terms of the unclosed PDF equations does not mean that mixing processes in the turbulent flames do not depend on combustion reactions. We proved this fact for the case of the laminar flamelet regime that persists with increasing turbulence up to extremely high levels due to the thin-reaction-zone nature of turbulent combustion.

Conclusion 2: The classical modeling strategy developed in the context of this paradigm, where the model PDF equations include the closed chemical source terms of the unclosed PDF equations and reaction-independent model mixing terms (to provide the closed-form description of the chemical effects caused by combustion reactions), is not suitable for modeling turbulent flames. This conclusion does not mean that the classical strategy has no areas for use. It is applicable to slow-reacting turbulent flows, where mixing processes are much faster than chemical transformations, and hence the use of reaction-independent mixing models is justified. In the case of PDF simulations of turbulent combustion incorporating detailed chemistry, the classical strategy is suitable for modeling chemical effects associated with relatively slow chemical reactions, such as after-burning of CO and thermal NO formation, while this strategy cannot be used to estimate prompt NO formation. It seems that the classical strategy can be suitable for modeling the so-called flameless or mild combustion (see, for example, Reference [23], where the EMST mixing model mentioned in the Section 1 was used in the PDF simulations of flameless combustion).

Conclusion 3: Considered in the paper, the alternative inverse and original joint direct/inverse modeling strategies in the PDF method allow the closed-form description of the chemical effects caused by the coupled reaction–diffusion process in the thin reaction zone assumed to be identical to the reaction zone in the laminar undisturbed turbulent flame. This is an extreme model simplification of the thin-reaction-zone nature of turbulent combustion in different burning regimes observed in experiments and confirmed by direct numerical simulations. This conclusion is valid when the width of the instantaneous reaction zone $\delta_r$ is much less than the size of the minimal turbulent eddies, $\delta_r << \eta$. With an increase in the level of turbulence, when the inequality becomes $\delta_r \geq \eta$, disturbances in the reaction zone will occur due to its stretching and curvature and the influence of small-scale turbulence. In this case, the closed-form description of chemical effects caused by fast combustion reactions is most likely impossible.

**Author Contributions:** Conceptualization, V.L.Z.; methodology, V.L.Z.; validation, V.L.Z.; formal analysis, V.L.Z.; investigation, V.L.Z.; resources, V.L.Z.; data curation, V.L.Z.; writing—original draft preparation, V.L.Z.; writing—review and editing, V.L.Z.; visualization, V.L.Z.; supervision, V.L.Z. All authors have read and agreed to the published version of the manuscript.

**Funding:** This research received no external funding.

**Data Availability Statement:** The data that supports the findings of this study are available within the article. Data sharing is not applicable to this article as no new data were created or analyzed in this study.

**Acknowledgments:** Thanks to Vladimir Frost, Forman Williams and Alexander Klimenko for helpful discussions of this topic.

**Conflicts of Interest:** The author declares that he has no conflict of interest.

## Nomenclature

| | |
|---|---|
| $(\overline{A}/A_0)$ | mean dimensionless area |
| $c$ | progress variable |
| $\hat{c}$ | averaged $c$ inside thickened flamelet |
| $D$ | molecular diffusion coefficient |
| DNS | direct numerical simulation |
| $D_t \sim u'L$ | turbulent diffusion coefficient |
| $Da = \tau_t/\tau_{ch}$ | Damköhler number |
| $f(n) = \|\nabla c\|(n)$ | gradient profile across flamelet |
| $H(x)$ | Heaviside step function |
| $L$ | integral scale of turbulence |
| $n$ | coordinate axis normal to flamelet |
| $p(c)$ | probability density function (PDF) |
| $\mathrm{Re}_t = u'L/\nu$ | turbulent Reynolds number |
| $S_L \sim (D/\tau_{ch})^{1/2}$ | speed of laminar flame |
| $S_{mt}$ | speed of thickened flamelet |
| $S_r$ | speed of reaction zone |
| $\vec{u}$ | flow velocity |
| $u' = (\overline{u'^2})^{1/2}$ | velocity fluctuation |
| $\overline{\vec{u}}_c$ | conditional mean velocity |
| $W$ | reaction rate per unit mass |
| **Greek Symbols** | |
| $\delta_L \sim (D\tau_{ch})^{1/2}$ | width of laminar flame |
| $\delta_{mt}$ | width of thickened flamelet |
| $\varepsilon$ | turbulent energy dissipation rate |
| $\overline{E}_c$ | conditional scalar dissipation rate |
| $\nu$ | kinematical viscosity coefficient |
| $\eta = L\mathrm{Re}_t^{-3/4}$ | Kolmogorov microscale |
| $\rho$ | gas density |
| $\tau_{ch} = D/S_L^2$ | combustion chemical time |
| $\tau_t = L/u'$ | turbulent time |
| $\omega_c$ | conditional mixing frequency. |
| **Subscripts and Superscripts** | |
| $\overline{a}$ | Reynolds averaging |
| $\tilde{a}$ | Favre averaging |
| $\hat{a}$ | local mean value inside flamelet |
| $b$ | refers to burned gas |
| $L$ | refers to laminar flamelet |
| $mt$ | refers to thickened flamelet |
| $r$ | refers to flamelet reaction zone |
| $u$ | refers to unburned gas |

## Appendix A. Derivation of the Unclosed PDF Equations Used in the Paper

The derivation is based upon the use of the Fourier transformation. The method was proposed by Vadim Kuznetsov in Reference [24] (see also Reference [25]).

Instantaneous $c-$ Equation (1) in the coordinate form is as follows

$$\rho\partial c/\partial t + \rho u_i \partial c/\partial x_i = \partial(\rho D \partial c/\partial x_i)/\partial x_i + \rho W(c) \tag{A1}$$

Introduce the characteristic function $\phi = \exp(i\alpha c)$. Mean value of $\varphi$ is the Fourier transformation of the PDF $p(c)$

$$\overline{\phi}(\alpha) = \int_0^1 \phi p(c)dc = \int_0^1 \exp(i\alpha c)p(c)dc \tag{A2}$$

Differentiate $\varphi$ by time and install in resulting expression $\partial c/\partial t$ from Equation (A1)

$$\partial\phi/\partial t = i\phi\{\alpha[-u_i\partial c/\partial x_i + (1/\rho)\partial(\rho D \partial c/\partial x_i)\partial x_i + W(c)]\} \tag{A3}$$

Using expression of a derivative of $\phi$ with coordinates $\partial\phi/\partial x_k = \exp(i\alpha c)\cdot i\alpha\cdot\partial c/\partial x_k$ we have the following

$$- i\phi\alpha u_k\partial c/\partial x_k = -u_k\partial\phi/\partial x_k$$

and

$$i\phi\alpha(1/\rho)\partial(\rho D\cdot\partial c/\partial x_k)\partial x_k =$$
$$= (1/\rho)\partial(\rho D\cdot\partial\phi/\partial x_k)/\partial x_k + \alpha^2 D\phi(\partial c/\partial x_k\cdot\partial c/\partial x_k).$$

Invoking the continuity equation $\partial\rho/\partial t + \partial\rho u_k/\partial x_k = 0$ and averaging results in an equation in terms of average characteristic function

$$\partial\overline{\rho\phi}/\partial t + \partial\overline{\rho u_k\phi}/\partial x_k = \partial(\overline{\rho D\cdot\partial\phi/\partial x_k})\partial x_k+$$
$$+\alpha^2\overline{\rho D\phi(\partial c/\partial x_k\cdot\partial c/\partial x_k)} + i\alpha\overline{\phi W}. \tag{A4}$$

For obtaining of the desired $p(c)-$ equation it is necessary to perform inverse Fourier transformation of all terms from Equation (A4), i.e., to multiply them by $\exp(-i\alpha c)$ and to integrate by $\alpha$. Since

$$\overline{\rho\phi} = \int \phi\rho(c)p(c)dc$$

the nonstationary terms is as it follows

$$[1/(2\pi)^2]\int \exp(-i\alpha c)(\partial\overline{\rho\phi}/\partial t)d\alpha = \partial[\rho(c)p(c)]/\partial t \tag{A5}$$

Since

$$\overline{\rho u_k\phi} = \iint \phi u_k\rho(c)p(u_k,c)dcdu_k = \int \phi(\overline{u}_k)_c\rho(c)p(c)dc$$

the convective terms is as it follows

$$[1/(2\pi)^2]\int \exp(-i\alpha c)\cdot\partial(\rho\overline{u_k\phi})/\partial x_k\cdot d\alpha = \partial[\rho p(c)(\overline{u}_k)_c]/\partial x_k, \tag{A6}$$

where $p(u_k, c)$ is a joint PDF, $(\overline{u}_k)_c$ is the conditional average velocity.

Similarly, transforming the diffusion term

$$D\partial\phi/\partial x_k = \overline{\partial\rho D\phi/\partial x_k} - \phi\overline{\partial\rho D/\partial x_k}$$

we have

$$[1/(2\pi)^2]\int \exp(-i\alpha c)\cdot\partial(\overline{\rho D\partial\phi/\partial x_k})/\partial x_k\cdot d\alpha = \partial(\rho D\cdot\partial p/\partial x_k)/\partial x_k \tag{A7}$$

The dissipative term can be presented as it follows

$$\overline{\rho\phi D(\partial c/\partial x_k\cdot\partial c/\partial x_k)} = \overline{\rho\phi E} = \iint \rho\phi Ep(E,c)dEdc = \int \rho(c)\phi\overline{E}_c p(c)dc$$

where $E = D(\partial c/\partial x_k \cdot \partial c/\partial x_k)$ and $\overline{E}_c = \overline{D(\partial c/\partial x_k \cdot \partial c/\partial x_k)}_c$ are instantaneous and conditional average dissipation rate of the progress variable $c$. The inverse Fourier transformation results

$$[1/(2\pi)^2]\int \exp(-i\alpha c)\overline{\rho\varphi E}d\alpha = \rho(c)\overline{E}_c(c)p(c)$$

and hence

$$[1/(2\pi)^2]\int \exp(-i\alpha c)\alpha^2 \overline{\varphi D(\partial c/\partial x_k \cdot \partial c/\partial x_k)})/\partial x_k \cdot d\alpha = -\partial^2(\rho(c)\overline{E}_c p(c))/\partial c^2. \quad \text{(A8)}$$

We notice that the second-order derivative with $c$ appears due to the multiplier $\alpha^2$. At transformation of the source term in Equation (A4) the multiplier $\alpha$ results in appearing of the first-order derivative with $c$

$$[1/(2\pi)^2]\int \exp(-i\alpha c)\overline{\varphi W(c)}d\alpha = -\partial[W(c)\rho(c)]/\partial c \quad \text{(A9)}$$

Installing Equations (A5)–(A9) in Equation (A4) we would have desired equation in terms of the PDF $p(c; \vec{x}, t)$ denoted by $p$ for brevity (Equation (2) in the main body of the text)

$$\partial(\rho p)/\partial t + \partial(\rho(\overline{u}_k)_c p)/\partial x_k - \partial(\rho D \partial p/\partial x_k)/\partial x_k + \\ \partial^2(\rho\overline{E}_c p)/\partial c^2 + \partial(\rho p W)/\partial c = 0. \quad \text{(A10)}$$

We ignored in Equation (2) the third molecular diffusion term that is justified at large Reynolds numbers.

The kinematical equation of the instantaneous laminar flame (3) with the left hand side in the coordinate form is as follows:

$$\rho\partial c/\partial t + \rho u_i \partial c/\partial x_i = \rho_u S_L|\nabla c| \quad \text{(A11)}$$

The comparison of Equations (A1) and (A11) leads to a formula

$$\rho_u S_L|\nabla c| = \partial(\rho D\partial c/\partial x_i)/\partial x_i + \rho W \quad \text{(A12)}$$

showing that the right-hand term in Equation (A11) is determined by the processes of molecular diffusion and chemical reaction that occur in an instantaneous laminar flame. The gradient of the progress variable can be presented as $|\nabla c| = dc/dn \geq 0$, where $n-$axis is normal to the laminar flame. Representing this expression as a function of the progress variable $|\nabla c| = f_L(c)$, for Equation (A12), we have intead of Equation (A4) the following equation

$$\partial\overline{\rho\varphi}/\partial t + \partial\overline{\rho u_k \varphi}/\partial x_k = i\alpha\rho_u S_L\overline{\varphi f_L/\rho} \quad \text{(A13)}$$

and instead of Equation (A9), we have the following equation

$$[1/(2\pi)^2]\int \exp(-i\alpha c)\rho_u S_L\overline{\varphi f_L/\rho}d\alpha = -\rho_u S_L\partial f_L/\partial c \quad \text{(A14)}$$

that lead to the desired unclosed PDF equation for the laminar flamelet regime as it follows

$$\partial(\rho p)/\partial t + \partial(\rho(\overline{u}_k)_c p)/\partial x_k + \rho_u S_L\partial(f_L p)/\partial c = 0. \quad \text{(A15)}$$

The Equations (A10) and (A15) can be represented in terms of the Favre PDF $\widetilde{p}(c; \vec{x}, t) = \overline{\rho p(c; \vec{x}, t)/\overline{\rho}}$ as follows:

$$\overline{\rho}\cdot\partial\widetilde{p}/\partial t + \overline{\rho}\widetilde{\vec{u}}\nabla\widetilde{p} + \nabla\cdot[(\overline{\rho\vec{u}''})_c\widetilde{p}] + \overline{\rho}\cdot\partial(\overline{E}_c\widetilde{p})/\partial c + \partial(W\overline{\rho}\widetilde{p})/\partial c = 0 \quad \text{(A16)}$$

and

$$\overline{\rho}\cdot\partial\widetilde{p}/\partial t + \overline{\rho}\widetilde{\vec{u}}\nabla\widetilde{p} + \nabla\cdot[(\overline{\rho\vec{u}''})_c\widetilde{p}] + \rho_u S_L\partial[(f_L/\rho)\overline{\rho}\widetilde{p}]/\partial c = 0, \quad \text{(A17)}$$

where $\vec{\rho}$ is the mean density, $\widetilde{\vec{u}} = \overline{\rho \vec{u}}/\overline{\rho}$ is the Favre-averaged velocity and $\vec{u}'' = \vec{u} - \widetilde{\vec{u}}$ is the Favre speed fluctuation, subscript $c$ means conditional averaging.

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
