# Peer review of "Conceptual Limitations of the Probability Density Function Method for Modeling Turbulent Premixed Combustion and Closed-Form Description of Chemical Reactions’ Effects"

_fluids, doi:10.3390/fluids6040142_

Round 1

Author Response

Review #1

Reply: The author thanks the reviewer for detail and many  critical remarks, which were taken into account in the revised version.  Author's comments and changes in the text are shown below.

Review of the paper “Limitations of the probability density function method for modeling turbulent premixed combustion and closed-form description of chemical reactions effects”. Submitted to Fluids Article number # : fluids-1061151 January 6, 2021

Recommendation: Accept with minor revisions.

Summary:

The present manuscript analyzes the potentialities and limitations of the probability density function method and possible misconceptions in its use for the modeling of turbulent premixed combustion. The topic is of interest and the analysis methodology is correct. Through detailed theoretical formulation derivations, the author shown that the classical modeling strategy is not suitable for modeling turbulent flames, and alternative inverse and original joint direct/inverse modeling strategies in the PDF method is proposed. In summary, this paper bring insights to the fundamental understandings of the use of PDF method to the turbulent modeling, and it is suggested to be published with some minor revisions.

Major concerns:

  • (1): In page 10, the manuscript mentioned that “...the reason of the overestimation is that the classical modeling strategy in the PDF method is, in fact, suitable for the distributed combustion mechanism only but not for the thin-reaction-zone combustion mechanism common to the turbulent flames in the differently classified regimes, which has been confirmed in recent years by optical experiments and direct numerical simulations...”. Comments: This is a strong comment, but the references are missing. The author is expected to show the papers support this conclusion.

The author’s comment:    

The references has been added: “optical experiments and direct numerical simulations [8]-[12],

Minor concerns:

  • (1): In page 17, the manuscript mentioned that “...In the next section we will consider the distributed preheat zone regime observed in [9] in premixed flames subjected to extreme levels of turbulence, and then, in the following section, we 18 will study the possibilities of the PDF method for modeling of the turbulent flame in the thickened flamelet regime, which is intermediate between the laminar flamelet regime (at relatively weak turbulence) and distributed preheat zone regimes (at very strong turbulence).....”. Comments: The sentence has many no-meaning repeating.

The author’s comment:    

This sentence has been modified as follows:

     In the next section we will consider the distributed preheat zone regime observed in [9] in premixed flames subjected to extreme levels of turbulence. Then, in the following section, we will examine the application of the PDF method for modeling of the turbulent flame in the thickened flamelet regime.

  • (2): In page 11, the manuscript mentioned that “...is not applicable for the description of the chemical effects associated with fast reactions (the heat release rate, prompt prompt nitrogen oxide (NO) formation) due to strong coupling of the molecular and these chemical processes in the thin instantaneous combustion zones observed in the experiments.....”. Comments: There is a typo.

The author’s comment:    The typo has been corrected.

  • (3): In page 14, the manuscript mentioned that “...This situations corresponds to a limiting case, when, in the known expressions for the width and speed of the laminar flame ....”. Comments: Subject-Verb Disagreement.

The author’s comment:    The typo has been corrected.

Reviewer 2 Report

The subject of the paper is interesting, but hard to focus on it I recommend major revision of the paper with the following observations: The references in the paper ahould be updated with more researsches in the past five years. There is a lack of images/figures which makes it harder to follow the article

Author Response

Review #2

Reply: The author thanks the reviewer for critical remarks. The author's comments and changes to the text are presented below.

Comments and Suggestions for Authors

The subject of the paper is interesting, but hard to focus on it I recommend major revision of the paper with the following observations:

The references in the paper should be updated with more researches in the past five years.

Author’s comment

We have added links to two relevant recent studies that are as follows:

[12] A.J. Aspden, M.S. Day, and J.B. Bell. "Towards the distributed burning regime in turbulent premixed flames." arXiv preprint arXiv:1806.09865 (2018).

[15]  Zhang, P., Xie, T., Kolla, H., Wang, H., Hawkes, E. R., Chen, J. H., & Wang, H. (2020).
„A priori analysis of a power-law mixing model for transported PDF model based on high
Karlovitz turbulent premixed DNS flames,” Proceedings of the Combustion Institute, (2020). https://doi.org/10.1016/j.proci.2020.06.183

There is a lack of images/figures which makes it harder to follow the article

Author’s comment

On the one hand, this remark is reasonable (the use of graphic illustrations is useful). But on the other hand, the work devoted to a critical analysis of the paradigm and concepts of PDF methods is addressed to readers working in the field of turbulent combustion modeling.

Reviewer 3 Report

The paper comes back on the common assumption about mixing models in PDF equations, which states that mixing is independent of chemistry. This is indeed not exactly true and it is clearly interesting to explain why and what are the possible implications. The paper then proposes alternative modelling approaches to overcome this shortcoming.

The paper truely addresses an interesting question, but requires major improvements before publishing, as suggested below.

First of all, as a general comment, the paper contains too many repetitions of the same idea (namely, that the assumption of independent mixing and chemistry is not always true). In additon, the introduction which bases the discussion on quoting other papers is not very useful and far too long. It is easy to explain why the assumption is not always true, using a combustion regime diagram and a scale analysis. I would sugget to skip all that part and come directly to the topic.

A second major weakness of the paper is that all affirmations are theoretical and not supported by quantitative data. Some experimental results are vaguely reported but this is not sufficient to be convincing. The problem with that approach is that, even if the assumption questioned in the paper is not true from a theoretical point of view, this is not enough to claim that it should not be used : all is a question of importance in comparison with other mechanisms.

The same comment holds for the proposed modelling approach : it is only theoretical and not tested on any  (even simple ) case. It should be at least illustrated.

Minor points and questions:

1- Page : the first name of Prof Borghi is Roland, not Ronald.

2- Page 9 : the author claims that their analysis and modeling could be applied to non-premixed combustion. However the modeling is based on the concept of flame speed, which does not exist for diffusion flames: how then could the model be applied?

3- Page 9 : the author claims the the analysis is performed for one-step chemistry, but this assumption never appears in the derivation. From my understanding the analysis applies to any type of chemsitry description.

4- Page 18 : in the case of high turbulence, the author states that the flame does not "have a clearly defined structure", but still model it "assuming that the structure of the laminar-type combustion reaction layer is known", which is contradictory. This should be clarified.

5- Page 18 : Eq. 9 is he same as Eq. 2 and does not need to be repeated.

6- Page 19 : the Heaviside function could be to abrupt for the change of model, could a relaxation approach be more robust?

7- Page 19 : how is the reaction layer determined ?

8- Page 21 : In the expression of f_mt, it should be the norm of the gradient of c.

9- Page 21 : how is P_tilde(clc_hat) determined?

10- Page 23 : Eq. 17 is a third repetition of Eq. 2 !

11- Page 23 : a double-cone burner is cited as an application but no results are shown or event mentioned. Why mentioning this case?

12- Appendix A can be skipped, it recalls the standard derivation of a PDF equation

13- Appendic B is useful but should be far more concise.

Author Response

Review #3

Reply: We thank the reviewer for their careful reading of the manuscript and their constructive remarks. We have taken the comments on board to improve and clarify the manuscript. Please find below a detailed point-by-point response to all comments.

The paper comes back on the common assumption about mixing models in PDF equations, which states that mixing is independent of chemistry. This is indeed not exactly true and it is clearly interesting to explain why and what are the possible implications. The paper then proposes alternative modelling approaches to overcome this shortcoming.

The paper truly addresses an interesting question, but requires major improvements before publishing, as suggested below.

First of all, as a general comment, the paper contains too many repetitions of the same idea (namely, that the assumption of independent mixing and chemistry is not always true). In additon, the introduction which bases the discussion on quoting other papers is not very useful and far too long. It is easy to explain why the assumption is not always true, using a combustion regime diagram and a scale analysis. I would sugget to skip all that part and come directly to the topic.

Author’s comment: 

  1. The assumption of independent mixing and chemistry is justified in the case of distributed combustion regime. The experimental and DNS results obtained in recent years show that the width instantaneous reaction zone even in the case of extremely strong turbulence is close to the width of the reaction zone of the laminar flame. Hence the assumption of independent mixing and chemistry is never justified in real flames. I  have added to the list of cited works an article  

[12] Aspden, A. J., M. S. Day, and J. B. Bell. "Towards the distributed burning regime in turbulent premixed flames." arXiv preprint arXiv:1806.09865 (2018)

which contains an attempt to produce in DNS distributed burning in extremely low Damköhler number lean methane and hydrogen flames ( for both fuels). “Turbulence has no survive deeper into the methane flame to disrupt the reaction zone”,  The results obtained in [12] show that the tendency towards a distributed burning regime took place only for hydrogen and with turbulent parameters that are not realized in experimental flames.

  1. I specifically cited the original formulation of the paradigm of the PDF modeling strategy which states that since a closed mixing term does not contain a chemical reaction, a model mixing term should not include a reaction. This implies the statement that the model PDF equations allow one to describe chemical effects in a closed form, which is discussed in the manuscript. The combustion regime diagram as atoolfor the analysis of the issue of the  mixing term modeling does not seem sutable. (We touch the question of the origin of the Borghi diagram in the manuscript).
  2. The intention of the author was that the reader, after looking over the somewhat extended introduction, understood the main statement of the article, and then decided whether to read the main text. It may turn out that for some part of the readers reading looking over the introduction and conclusions will be sufficient.

A second major weakness of the paper is that all affirmations are theoretical and not supported by quantitative data. Some experimental results are vaguely reported but this is not sufficient to be convincing. The problem with that approach is that, even if the assumption questioned in the paper is not true from a theoretical point of view, this is not enough to claim that it should not be used : all is a question of importance in comparison with other mechanisms.

The same comment holds for the proposed modelling approach : it is only theoretical and not tested on any  (even simple ) case. It should be at least illustrated

Author’s comment: 

The author addresses a fundamental  issue concerning the opportunity of the closed-form description of the chemical effects caused by combustion reactions in the context of the PDF method, a theoretical consideration.  Recent experimental and DNS data show that the distributed combustion regime that is the condition for applying the standard PDF modeling strategy is not realized in real turbulent flames.

 The paper does not contain the clam the standard modeling strategy should not be used.  Quotations:  “At the same time, the standard PDF method is suitable for modeling of turbulent non-premixed flames in the mixing-controlled combustion regime and estimation of the chemical effects caused by slow reactions (the thermal NO formation and slow carbon monoxide CO oxidation)” and “the direct modeling strategy is applicable to slow-reacting turbulent flows and for modeling chemical effects in turbulent flames associated with slow reactions”. The clain is that the standard (direct) modeling strategy is not sutable for simulations of the turbulent premixed flame due to  the thin-reaction-zone nature of turbulent combustion

Minor points and questions:

1- Page : the first name of Prof Borghi is Roland, not Ronald.

Author’s comment:  This mistake has been corrected:

2- Page 9 : the author claims that their analysis and modeling could be applied to non-premixed combustion. However the modeling is based on the concept of flame speed, which does not exist for diffusion flames: how then could the model be applied?

Author’s comment:

In order to avoid confusion, we have replaced

“We restrict our analysis to turbulent premixed combustion although the main conclusions are applicable also to non-premixed combustion.” 

by

Although we have limited our analysis to turbulent premixed combustion, the conclusion of this study about the inapplicability of the classical modeling strategy for assessing the chemical effects caused by fast combustion reactions (in particular the formation of prompt nitrogen oxide NO) also refers to non-premixed combustion. At the same time, the standard PDF method is suitable for modeling of turbulent non-premixed flames in the mixing-controlled combustion regime and estimation of the chemical effects caused by slow reactions (the thermal NO formation and slow carbon monoxide CO oxidation).

. We did not analyze the application of PDF methods to non-premixed combustion and therefore did not consider models for assessing the chemical effects of the fast reactions that inevitably  would be different. 

3- Page 9 : the author claims the the analysis is performed for one-step chemistry, but this assumption never appears in the derivation. From my understanding the analysis applies to any type of chemistry description.

Author’s comment: You are right.

“Though we consider the case of a one-step combustion reaction, the main conclusions are also applicable to the PDF modeling of combustion using detailed chemical kinetics”

has been replaced by

“Although for simplicity we consider the PDF equations corresponding to the case of one-step reaction, the main conclusions are also applicable to the PDF modeling of combustion using detailed chemical kinetics.”

4- Page 18 : in the case of high turbulence, the author states that the flame does not "have a clearly defined structure", but still model it "assuming that the structure of the laminar-type combustion reaction layer is known", which is contradictory. This should be clarified.

You are right. This issue was clarified as follows:

Old: This means that “the major issue with mixing models in PDF methods” is not justified in this case. At the same time, the instantaneous flame does not have a clearly defined structure, which is a condition for the applicability of the inverse PDF modeling strategy.

Modified: Due to the thin-reaction-zone nature, “the major issue with mixing models in PDF methods” is not justified. At the same time, in contrast to premixed turbulent combustion in the laminar or thickened flamelet regimes, in the case of the distributed preheat zone, the turbulent flame does not have a clearly defined sheet-type structure, which is a condition for the applicability of the inverse PDF modeling strategy.

5- PagIl telefono funziona, ma crepitio  molto forte, quindi la comunicazione è impossibile.e 18 : Eq. 9 is he same as Eq. 2 and does not need to be repeated.

Author’s comment:  This is not entirely true. They are two different forms of the same equations.  Eq.2 for the PDF p(c;x,t) and Eq. (9) for the Favre PDF were derived in the Appendix A:  Eq. (A10) and Eq. (A16). In the following text, we used Eq. (9).

6- Page 19: the Heaviside function could be to abrupt for the change of model, could a relaxation approach be more robust?

Author’s comment:  

We considered a fundamental point. Smoothing the boundary between models may be necessary in numerical simulations.

7- Page 19 : how is the reaction layer determined ?

The reaction rate defined in the cited paper [?]  and some other works optically by analysis of the OH and CH20 images. In theoretical studies, the reaction layer is the zone where most of the heat is released.

Author’s comment:  

8- Page 21 : In the expression of f_mt, it should be the norm of the gradient of c.

Author’s comment:  The remark is correct: the gradient symbol was mistakenly omitted

9- Page 21 : how is P_tilde(clc_hat) determined?

Author’s comment:  This Favre PDF is defined in the Appendix A. For the reader’s convenience I have edded in the main body the formula defining this Favre PDF just before Eq. (9).

10- Page 23 : Eq. 17 is a third repetition of Eq. 2 !

Author’s comment:  Eq. (17) is a repetition of Eq. (9). (as we mentioned above, Eq. (2) and (9) are different).

The equation (17) has been removed from the text and replaced with reference Eq. (9).

11- Page 23 : a double-cone burner is cited as an application but no results are shown or event mentioned. Why mentioning this case?

Author’s comment:  We have excluded the mention of a double industrial burner here since we are addressing this issue in Appendix B.

12- Appendix A can be skipped, it recalls the standard derivation of a PDF equation

Author’s comment:

This proposed in 1967 by Kuznetzov’s method of the rigorous mathematic derivation of the PDF equation is not the standard one. Usual methods used by many researchers originated by Lundgren in the paper  Lundgren, T. S. (1969). Model equation for nonhomogeneous turbulence. The Physics of Fluids12(3), 485-497starts from the presentation for the PDFs as expressions of Dirac delta functions.  The use of the tool of generalized functions is not inevitable in this problem. Using in the Appendix this effective and not very well-known method, we also popularize it among researchers.

13- Appendix B is useful but should be far more concise.

Author’s comment:

Currently, the main focus is on the technology: modeling, numerical simulations, comparison with experiments, applications. A sufficiently detailed analysis of the discussions on the nature of turbulent combustion in which prominent scientists took part seems useful. Moreover, some old concepts that were not confirmed in subsequent years, are still used in practical modeling, in particular the concept of a distributed combustion regime in flows with strong turbulence,

Reviewer 4 Report

Review of a manuscript entitled
“Limitations of the probability density function method for modeling turbulent premixed combustion and closed-form description of chemical reactions effects ”
by Vladimir Zimont
submitted to Fluids (ISSN 2311-5521)

The manuscript has the aim to analyze the possibilities of probability density function (PDF) methods for the closed-form description of combustion chemical effects in turbulent premixed flames. The reason for this is the strong dependence of mixing on combustion reactions due to the nature of the thin reaction zone of turbulent combustion confirmed by optical studies and direct numerical simulations. In this case, the chemical effect is determined by coupled reaction-diffusion processes that take place in thin zones of instantaneous combustion.
The possible alternative modeling strategies in the PDF method, considered in this study, would allow to describe the chemical effects in a closed form. This is possible only in a hypothetical case in which instant combustion occurs in reaction zones identical to the reaction zone of the undisturbed laminar flame. For turbulent combustion in the laminar flame regime, we use an inverse modeling strategy where the PDF model directly contains the laminar flame characteristics. For turbulent combustion in the distributed preheating zone regime, an original joint direct / reverse modeling strategy is offered.

The introduction describes in an exhaustive way the analyzed problem and the available literature. In this way the reader has a global vision of the limits of the models already developed and highlights the need for a model with a different strategy.

Subsequently, the two models built are described in a clear and exhaustive way, both from a mathematical and a descriptive point of view.

In conclusion, the study was conducted to analyze the potential and limitations of the PDF method and possible misunderstandings in its use for modeling turbulent premixed combustion. This work led to the conclusion that the absence of reaction rates in the unknown mixing terms of the non-closed PDF equations does not mean that the mixing processes in turbulent flame are not dependent on combustion reactions. This fact has been demonstrated for the case of the laminar flame regime which persists with increasing turbulence to extremely high levels due to the nature of the thin reaction zone of turbulent combustion.
Another very interesting conclusion is the fact that the classical modeling strategy developed in the context of this paradigm, where the PDF model equations include the closed chemical origin terms of the non-closed PDF equations and the reaction-independent model mixing terms it is not suitable for modeling turbulent flames. It is therefore applicable to slow-reacting turbulent flows, where mixing processes are much faster than chemical transformations, and therefore the use of reaction-independent mixing models is justified. The classical strategy appears to be suitable for modeling so-called flameless or mild combustion.

The alternative direct / reverse and original joint modeling strategies in the PDF method allow for the closed form description of the chemical effects caused by the coupled reaction-diffusion process in the thin reaction zone which is assumed to be identical to the reaction zone in the undisturbed turbulent laminar flame. It is an extremely simplified model of the nature of the thin reaction zone of turbulent combustion in different combustion regimes observed in the experiments and confirmed by direct numerical simulations.

In conclusion, this work is well written and very interesting to read, with an interesting and important objective in the field of both premixed and non-premixed combustion. I recommend publishing this article, without major changes.

Author Response

Review #4

Reply: We thank the reviewer for their careful reading of the manuscript and their positive comments.  

Review of a manuscript entitled 
“Limitations of the probability density function method for modeling turbulent premixed combustion and closed-form description of chemical reactions effects ”
by Vladimir Zimont
submitted to Fluids (ISSN 2311-5521)

The manuscript has the aim to analyze the possibilities of probability density function (PDF) methods for the closed-form description of combustion chemical effects in turbulent premixed flames. The reason for this is the strong dependence of mixing on combustion reactions due to the nature of the thin reaction zone of turbulent combustion confirmed by optical studies and direct numerical simulations. In this case, the chemical effect is determined by coupled reaction-diffusion processes that take place in thin zones of instantaneous combustion.
The possible alternative modeling strategies in the PDF method, considered in this study, would allow to describe the chemical effects in a closed form. This is possible only in a hypothetical case in which instant combustion occurs in reaction zones identical to the reaction zone of the undisturbed laminar flame. For turbulent combustion in the laminar flame regime, we use an inverse modeling strategy where the PDF model directly contains the laminar flame characteristics. For turbulent combustion in the distributed preheating zone regime, an original joint direct / reverse modeling strategy is offered.

The introduction describes in an exhaustive way the analyzed problem and the available literature. In this way the reader has a global vision of the limits of the models already developed and highlights the need for a model with a different strategy.

Subsequently, the two models built are described in a clear and exhaustive way, both from a mathematical and a descriptive point of view.

In conclusion, the study was conducted to analyze the potential and limitations of the PDF method and possible misunderstandings in its use for modeling turbulent premixed combustion. This work led to the conclusion that the absence of reaction rates in the unknown mixing terms of the non-closed PDF equations does not mean that the mixing processes in turbulent flame are not dependent on combustion reactions. This fact has been demonstrated for the case of the laminar flame regime which persists with increasing turbulence to extremely high levels due to the nature of the thin reaction zone of turbulent combustion.
Another very interesting conclusion is the fact that the classical modeling strategy developed in the context of this paradigm, where the PDF model equations include the closed chemical origin terms of the non-closed PDF equations and the reaction-independent model mixing terms it is not suitable for modeling turbulent flames. It is therefore applicable to slow-reacting turbulent flows, where mixing processes are much faster than chemical transformations, and therefore the use of reaction-independent mixing models is justified. The classical strategy appears to be suitable for modeling so-called flameless or mild combustion.

The alternative direct / reverse and original joint modeling strategies in the PDF method allow for the closed form description of the chemical effects caused by the coupled reaction-diffusion process in the thin reaction zone which is assumed to be identical to the reaction zone in the undisturbed turbulent laminar flame. It is an extremely simplified model of the nature of the thin reaction zone of turbulent combustion in different combustion regimes observed in the experiments and confirmed by direct numerical simulations.

In conclusion, this work is well written and very interesting to read, with an interesting and important objective in the field of both premixed and non-premixed combustion. I recommend publishing this article, without major changes.

Round 2

Reviewer 2 Report

The paper isn't improved according to the previous observations

E.g. old references
lack of visual data (figures, tables)

I recommend to reanalyse the entire paper and rewrite it according to the recent results in literature

I recommend to reject the paper

Author Response

Reviewer: “The paper isn't improved according to the previous observations”

Author’s comment

This is not true. Answering the suggestion

“The references in the paper should be updated with more researches in the past five years”

The author has updated to References of the first modified version two citations

[12] A.J. Aspden, M.S. Day, and J.B. Bell. "Towards the distributed burning regime in turbulent premixed flames."  arXiv preprint arXiv:1806.09865  (2018).

[15] Zhang, P., Xie, T., Kolla, H., Wang, H., Hawkes, E. R., Chen, J. H., & Wang, H. (2020).
„A priori analysis of a power-law mixing model for transported PDF model based on high
Karlovitz turbulent premixed DNS flames,” Proceedings of the Combustion Institute, (2020)

and additionally to References of the second modified version issued just now

W. Zhang, J. Wang, W. Lin, G. Li, Z. Hu, M. Zhang, and Z. Huang, Effect of hydrogen enrichment on flame broadening of turbulent premixed flames in thin reaction regime. International Journal of Hydrogen Energy, 46(1), 1210-1218 (2021).

Reviewer: “E.g. old references”

Author’s comment

The “old references” [2]-[4], [16]-[18] refer to the classical articles in the field of PDF methods discussed in our manuscript. These papers are discussed in the manuscript.

Reviewer: “lack of visual data (figures, tables)”

Author’s comment

The principles and paradigms of the PDF method for turbulent combustion were formulated in [2]-[4] without using figures and tables. Our analysis of the PDF method in the context of recent experimental and DNS data on the structure of instantaneous combustion zones also does not require the use of graphs and tables. (they are necessary when presenting the results of numerical modeling).

Reviewer:I “recommend to reanalyze the entire paper and rewrite it according to the recent results in literature”

Author’s comment

The principles and paradigm of the PDF method were formulated in the "old references" in the framework of the concept of the distributed combustion reactions in the developed turbulent flames. This issue did not consider in the recent literature mainly devoted to practical modeling. The need for a theoretical reanalysis of this fundamental issue arose in connection with the  experimental and DNS results obtained    during the last five years using modern optical and numerical methods. that we concider in the manuscript. These results showed that the basic assumption of the classical PDF modeling strategy about the distributed combustion reaction is not fulfilled in real flames even in the case when they are subjected to extreme levels of turbulence.

I Reviewer: “I recommend to reject the paper”

Author’s comment

I would agree with this recommendation if the reviewer showed that our analysis of this issue and the proposed modification of the PDF modeling strategy taking into account the real nature of developed turbulent flames are erroneous. But the author did not find in the review a statement   about the fallacy or doubtfulness of our  results.

Reviewer 3 Report

The authors have answered the questions of my first review in a convincing way. However they do not consider to follow my suggestion of shortening some parts and being more concise in some explanations. I believe that this weakens the paper and makes it inadequate for publication. For that reason, I do not change my original rating, although I think that the paper has a true scientific quality.

Author Response

“The authors have answered the questions of my first review in a convincing way”.

Author's comment:

All questions and comments were substantiated and therefore taken into account.

“However they do not consider to follow my suggestion of shortening some parts and being more concise in some explanations. I believe that this weakens the paper and makes it inadequate for publication. For that reason, I do not change my original rating, although I think that the paper has a true scientific quality.”

Author's comment:

The text has been reduced from 11,828 words to 9,128 words, which means that the previous text was one-third more than the last one. The second appendix has been omitted. The introduction has been substantially shortened. Several changes have been made to the text of the manuscript to make it clearer and more concise. A brief review of the recent experimental and DNS results obtained in [11] - [18], which formed the basis of our PDF approach, were separated into a separate section.

The author hopes that the manuscript modified in this way will be suitable for publication.

Round 3

Reviewer 2 Report

The paper has been partially improved. I would recommend to publish the paper.

Reviewer 3 Report

Although I am not much comfortable with the way the authors present their work, as I already described in my first review, this is probably not an argument for a review.

Therefore I suggest to accept the paper.